# Relationship between C-reactive protein-albumin ratio and prognosis in pediatric patients with severe burns

Serap Samut Bülbül[1], Alper Ceylan[2], Metin Ocak[3], Murat Güzel[3], Selim Görgün[4]*

1 Faculty of Medicine, Department of Pediatric Surgery, Samsun University, Samsun, Turkey,
2 Department of General Surgery, Samsun Training and Research Hospital, Samsun, Turkey, 3 Faculty of Medicine, Department of Emergency Medicine, Samsun University, Samsun, Turkey, 4 Department of Microbiology and Clinical Microbiology, Samsun Traning and Research Hospital, Samsun, Turkey

* selimgorgun55@gmail.com

## Abstract

C-reactive protein (CRP) is a widely requested acute-phase protein. This study aimed to investigate the relationship between albumin, CRP, and CRP-albumin ratio values and hospital length of stay (LOS) in children with severe burns. A total of 150 pediatric patients who were treated in our hospital's burns clinic or burns intensive care unit due to third- and fourth-degree burns between January 2019 and September 2023 were included in the study. The mean CRP-albumin ratio was significantly lower in those hospitalized for 10 days or less than that in those hospitalized for a longer period (p < 0.001). The rates of patients with high CRP-albumin ratio in those hospitalized for 21–30 days and more than 40 days were found to be significantly higher than the other groups (p < 0.001). Hospital LOS was positively and significantly related to CRP-albumin ratios (p < 0.001; r = 0.529). In children with severe burns, low albumin levels, high CRP levels, and high CRP-albumin ratios at hospital admission may be related to clinical presentation and hospital LOS.

## Introduction

Severe burns in children are one of the causes of morbidity and mortality. In pediatric patients exposed to severe burns, many factors, especially dehydration and infection at the wound site, delay healing and cause many complications, including mortality. After burn exposure, a significant inflammatory process begins in the body, and inflammatory markers provide important information about the patient's clinical condition and clinical course [1,2].

C-reactive protein (CRP), an acute-phase reactant, increases with the development of any inflammation in patients. It has been shown that CRP values are associated with the clinical course of many diseases and decrease to normal levels with treatment [3,4]. Albumin is the primary protein that maintains plasma colloid osmotic

**Data availability statement:** The data is available at the following link: (http://datadryad.org/share/kdo7wyUS7qHLXqwy275fw07JpT42O-0Jy_gablTc3JkI).

**Funding:** The author(s) received no specific funding for this work.

**Competing interests:** The authors have declared that no competing interests exist.

pressure. Albumin, which has an important role in regulating plasma osmotic pressure, serves as a carrier for many endogenous and exogenous compounds and as a buffer molecule for acid and base balance, and it also has important anti-inflammatory antioxidative properties. In patients with burns, significant tissue and protein loss occurs and the patient develops severe edema. In this case, a significant change is observed in serum albumin levels, a plasma protein [5–8]. Studies have shown that albumin levels decrease in some critical diseases and that this decrease provides information in terms of clinical course, morbidity, and mortality [5–7]. In recent years, it has been reported that the CRP-albumin ratio is informative for the clinical course of some infections, cancerous and non-cancerous diseases, and that high CRP-albumin ratios may indicate a worse prognosis in these patients due to high CRP and low serum albumin levels [9–12].

This study aimed to investigate the relationship between albumin, CRP, and CRP-albumin ratio values and the hospital length of stay (LOS) in children with severe third-degree and above burns.

## Materials and methods

### Ethical approval and informed consent

The study was conducted as a retrospective cohort study. Samsun University Clinical Trials Ethics Committee approved the study (Date: 12/10/2023, Protocol no: SÜKAEK-2023 19/12). Due to the retrospective nature of the study, informed consent was waived because the study data were collected in an anonymous manner.

### Patients and data

A total of 150 pediatric patients with third and fourth-degree burns who were evaluated in the emergency department of our hospital between January 1st, 2019, and September 30th, 2023, and who were admitted to the burns clinic or burns intensive care unit (ICU) with an indication for hospitalization were included in the study. The authors accessed the data for research purposes from October 20th to November 25th, 2023. Patients who were hospitalized and discharged at the day of admission were excluded from the study. Patients with burns with immunodeficiency, a history of autoimmune disease, and active oncologic and rheumatologic diseases were not included in the study. The laboratory results of the patients' first hospital admission were obtained by scanning the records retrospectively from the hospital automation system. Patients aged 18 years and over and those who died during burns treatment were not included in the study due to their very low number. Patients with thermal and electrical burns were included in the study. A patient who died within the first 24 hours was excluded from the study because of accompanying inhalation burns.

Patients with albumin values below 35 g/L were considered to have hypoalbuminemia. Albumin replacement therapy was administered in the clinical follow-up of patients with albumin values below 25 g/L. Grafting was performed during the follow-up of patients with deep burns if epithelialization was insufficient after debridement.

For CRP-albumin, a threshold value of 1.215 was determined in receiver operating characteristic (ROC) analysis, values below this were considered low, and above this were high.

## Statistical analysis

The sample size in the study was calculated through power analysis using the G*Power software (version 3.1.9.6, Franz Faul, Universitat Kiel, Germany). Effect size 0.6; Type 1 error was taken as 0.05 and test power as 0.95, and the total sample size was calculated as 116.

All statistical analyses in the study were performed using the SPSS 25.0 software (IBM SPSS, Chicago, IL, USA). Descriptive data are given as numbers and percentages. Comparisons between groups in terms of categorical variables were made using the Chi-square test and Fisher's exact test. The Kolmogorov-Smirnov test was used to verify whether continuous variables were normally distributed. Differences between groups in terms of continuous variables were analyzed using the Mann-Whitney U test and the Kruskal-Wallis test. The relationship between continuous variables was evaluated using Spearman's correlation analysis. The ability of the CRP-albumin ratio to predict hospital LOS was analyzed using ROC analysis. The results were evaluated within the 95% confidence interval and p-values of $< 0.05$ were considered significant.

## Results and discussion

The mean age of the patients was $3.7 \pm 4.2$ years (range: 4 months-17 years) and 88 (58.7%) were male.

The mean serum albumin level in patients hospitalized for more than 20 days were significantly lower than in those hospitalized for 20 days or less ($p < 0.001$). The mean CRP level in those hospitalized for 10 days or less was significantly lower than in those hospitalized for longer, and the mean CRP level in those hospitalized for 21–30 days and more than 40 days was significantly higher than the other groups ($p < 0.001$). The mean CRP-albumin ratio in those hospitalized for 10 days or less was significantly lower than in those hospitalized for longer, and the mean CRP level in those hospitalized for 21–30 days and more than 40 days was significantly higher than the other groups ($p < 0.001$) (**Table 1**).

The rates of patients with high CRP-albumin ratio in those hospitalized for 21–30 days and more than 40 days were found to be significantly higher than the other groups ($p < 0.001$) (**Table 2**).

In the correlation analyses, the hospital LOS was inversely and significantly related to albumin levels at admission ($p < 0.001$; $r = -0.333$), and was positively and significantly related to CRP levels ($p < 0.001$; $r = 0.495$) and CRP-albumin ratios ($p < 0.001$; $r = 0.529$) (**Table 3**).

In the ROC analysis, the sensitivity and specificity values in predicting hospital LOS of more than 20 days were found as 89.6% and 58.8%, respectively, for a threshold value of 1.215 in the CRP-albumin ratio at the time of admission (area under curve $= 0.83$; $p < 0.001$; lower bound $= 0.765$; upper bound $= 0.896$) (**Fig 1**).

**Table 1. Mean albumin, CRP, and CRP-albumin ratio values according to hospitalization duration.**

| | Albumin (g/L) | | CRP (mg/L) | | CRP/Albumin | |
| --- | --- | --- | --- | --- | --- | --- |
| | **Mean** | **SD** | **Mean** | **SD** | **Mean** | **SD** |
| Hospital stay | | | | | | |
| 1-10 days | 34.21 | 5.18 | 31.37 | 27.06 | 0.99 | 0.97 |
| 11-20 days | 32.82 | 4.67 | 58.87 | 46.34 | 1.84 | 1.45 |
| 21-30 days | 27.81 | 6.68 | 86.49 | 45.98 | 3.56 | 2.09 |
| 31-40 days | 26.67 | 6.93 | 69.19 | 33.45 | 2.62 | 1.08 |
| >40 days | 27.26 | 9.25 | 115.35 | 59.84 | 4.78 | 3.06 |
| p | **<0.001** | | **<0.001** | | **<0.001** | |

*Note:Kruskal-Wallis Test was used. SD: Standard deviation, CRP: C-reactive protein.*

**Table 2. Distribution of CRP-albumin ratio levels according to hospitalization duration.**

| | CRP/Albumin | | | | | |
| | Normal | | High | | Total | P |
| | n | % | n | % | n | |
|---|---|---|---|---|---|---|
| Hospital stay | | | | | | **<0.001** |
| 1-10 days | 52 | 100.0 | 0 | 0.0 | 52 | |
| 11-20 days | 46 | 92.0 | 4 | 8.0 | 50 | |
| 21-30 days | 11 | 55.0 | 9 | 45.0 | 20 | |
| 31-40 days | 10 | 90.9 | 1 | 9.1 | 11 | |
| >40 days | 10 | 58.8 | 7 | 41.2 | 17 | |
| Total | 129 | 86.0 | 21 | 14.0 | 150 | |

*Note: Chi-square and Fisher's exact test were used. CRP: C-reactive protein.*

**Table 3. Correlation analysis between patients' hospital stay, albumin, CRP, and CRP/Albumin values.**

| | | Age | Hospital stay |
|---|---|---|---|
| Hospital stay | r | 0.014 | |
| | p | 0.865 | |
| Albumin | r | 0.068 | -.333 |
| | p | 0.410 | **<0.001** |
| CRP | r | 0.016 | .495 |
| | p | 0.847 | **<0.001** |
| CRP/Albumin | r | 0.003 | .529 |
| | p | 0.968 | **<0.001** |

*Note: Pearson's correlation analysis was used. CRP: C-reactive protein*

Complications leading to death may develop in children exposed to severe burns [1,2]. In our study, it was concluded that the CRP-albumin ratio in pediatric patients with burns could guide physicians at a moderate level about the prognosis of the patients.

It has been reported that low serum albumin levels are an indicator of poor prognosis in patients with sepsis [8,13] and surgical patients [14]. In our study, the mean serum albumin level in patients hospitalized for more than 20 days were significantly lower than in those hospitalized for 20 days or less. In the correlation analysis, it was determined that the hospital LOS was negatively and significantly related to the albumin levels at hospital admission. These findings show that the albumin level at hospital admission may be informative in predicting the clinical picture and hospital LOS in children with severe burns. Accordingly, children with low albumin levels take longer to recover.

CRP is produced in the liver, macrophages, and adipose tissue. CRP levels begin to increase in the early stages of the inflammatory process. CRP levels may change in proportion to the clinical course and progress of treatment and is used in treatment follow-up [3,4,15]. In our study, the mean CRP level in those hospitalized for 10 days or less was significantly lower than in those hospitalized for longer, and the mean CRP level in those hospitalized for 21–30 days and more than 40 days was significantly higher than the other groups. In the correlation analysis, it was seen that hospital LOS was positively and significantly related to the CRP levels at hospital admission. All these findings show that CRP levels at hospital admission may be informative about the clinical picture and recovery time in pediatric patients with severe burns.

In recent years, inspired by the fact that elevated CRP and decreased serum albumin levels are informative in the clinical course of some diseases, the clinical value of the CRP-albumin ratio has begun to be investigated [9–12,16–27].

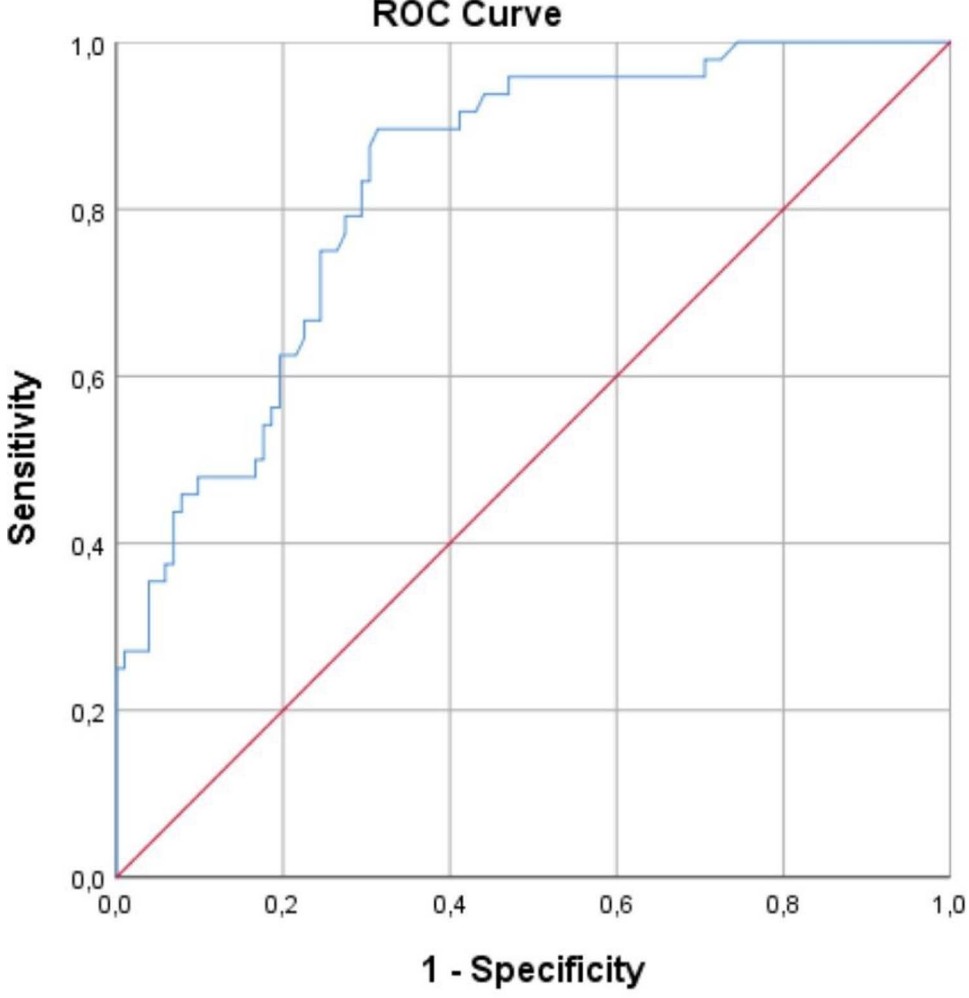

**Fig 1. Evaluation of the relationship between patients' CRP-albumin ratio values and hospital stay using ROC analysis.**

It has been reported that the CRP-albumin ratio provides important information about prognosis and that a high CRP-albumin ratio can be an indicator of poor prognosis in many situations such as sepsis [9,16,17], coronavirus disease-2019 [12,18,19], tuberculosis [20], pancreatitis [21,22], coronary diseases [11,23], cancer [24–26], and surgery [27]. In our study, the mean CRP-albumin ratio in those hospitalized for 10 days or less was significantly lower than in those hospitalized for longer, and the mean CRP level in those hospitalized for 21–30 days and more than 40 days was significantly higher than the other groups. In the correlation analysis, it was determined that hospital LOS was positively and significantly related to the CRP-albumin ratios at hospital admission. In addition, in the ROC analysis, the sensitivity and specificity values in predicting hospital LOS of more than 20 days were found as 89.6% and 58.8%, respectively, for a threshold value of 1.215 in the CRP-albumin ratio at the time of admission. All those findings show that CRP to albumin rate can be related with the duration of hospital stay.

There were some limitations in our study. Although the study covered a wide period of 4.5 years, pediatric patients who died could not be included in the study because the number was very low, thus whether values such as CRP-albumin ratio

were informative in predicting mortality could not be analyzed. Three patients with burns died in the first 24 hours during the study period (one burn caused in a house fire and two by lightning strikes, respectively). Three patients died after the first 24 hours (one burn caused by a flame and two caused by hot water, respectively). Because only three patients died in the burn center after 24 hours, this was not considered sufficient for statistical evaluation, accordingly, the effect on mortality could not be examined. In the study, patients were not evaluated according to burn area location, which could affect burn healing, complications, and hospital stay. Conducting the study on patients treated at a single hospital may limit the generalizability of the findings to other regions, health systems, or patient populations. The threshold value for the CRP-to-albumin ratio was derived from the study's ROC analysis. This may lead to overfitting and the need for external validation in larger, independent cohorts to confirm the applicability of the threshold. The retrospective nature of the study may have introduced potential biases such as incomplete data collection or variability in clinical practices during the study period. Only CRP, albumin, and CRP-albumin ratios were analyzed at admission. Longitudinal changes in these markers may provide more detailed information about their relationship with clinical course and outcome. Although the sample size calculation provides sufficient power, the exclusion of some subgroups (e.g., those with certain comorbidities) may limit the scope of the study.

## Conclusions

The findings obtained from our study showed that low albumin values, high CRP values, and high CRP-albumin ratios at admission to the hospital in children with severe burns may be related to the clinical picture and hospital LOS, and that the CRP-albumin ratio has moderate sensitivity in predicting the hospital LOS.

## Acknowledgments

We would like to thank the burns ward and intensive care staff.

## Author contributions

**Conceptualization:** Serap Samut Bülbül, Murat Güzel, Selim Görgün.

**Data curation:** Serap Samut Bülbül, Alper Ceylan, Metin Ocak, Selim Görgün.

**Formal analysis:** Metin Ocak, Selim Görgün.

**Funding acquisition:** Serap Samut Bülbül.

**Investigation:** Serap Samut Bülbül, Selim Görgün.

**Methodology:** Alper Ceylan, Metin Ocak, Murat Güzel, Selim Görgün.

**Project administration:** Metin Ocak.

**Resources:** Serap Samut Bülbül, Alper Ceylan.

**Supervision:** Alper Ceylan, Metin Ocak, Murat Güzel.

**Visualization:** Alper Ceylan, Murat Güzel.

**Writing – original draft:** Serap Samut Bülbül, Selim Görgün.

**Writing – review & editing:** Alper Ceylan, Metin Ocak, Murat Güzel.

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
