## [Decision Letter · Decision Letter 0]

13 Jan 2025

PONE-D-24-38904Relationship Between C-reactive protein-Albumin Ratio and Prognosis in Pediatric Severe Burn PatientsPLOS ONE

Dear Dr. Görgün,

Thank you for submitting your manuscript to PLOS ONE. After careful consideration, we feel that it has merit but does not fully meet PLOS ONE’s publication criteria as it currently stands. Therefore, we invite you to submit a revised version of the manuscript that addresses the points raised during the review process.

We look forward to receiving your revised manuscript.

Kind regards,

Keiko Hosohata, Ph.D.

Academic Editor

PLOS ONE

Journal Requirements:

2. In the online submission form, you indicated that [Data related to the manuscript will be made available by the authors upon request, provided that appropriate conditions are met].

[The authors have declared that no competing interests exist.

I have read the journal's policy and the authors of this manuscript have the following competing interests: [insert competing interests here].

Please confirm that this does not alter your adherence to all PLOS ONE policies on sharing data and materials, by including the following statement: ""This does not alter our adherence to PLOS ONE policies on sharing data and materials.” (as detailed online in our guide for authors http://journals.plos.org/plosone/s/competing-interests ). If there are restrictions on sharing of data and/or materials, please state these. Please note that we cannot proceed with consideration of your article until this information has been declared.

5. Please ensure that you include a title page within your main document. You should list all authors and all affiliations as per our author instructions and clearly indicate the corresponding author.

6. Please include captions for your Supporting Information files at the end of your manuscript, and update any in-text citations to match accordingly. Please see our Supporting Information guidelines for more information: http://journals.plos.org/plosone/s/supporting-information .

Reviewers' comments:

Reviewer's Responses to Questions

**Comments to the Author**

1. Is the manuscript technically sound, and do the data support the conclusions?

Reviewer #1: Partly

Reviewer #2: Yes

Reviewer #3: Yes

2. Has the statistical analysis been performed appropriately and rigorously? 

Reviewer #1: I Don't Know

Reviewer #2: Yes

Reviewer #3: Yes

3. Have the authors made all data underlying the findings in their manuscript fully available?

Reviewer #1: No

Reviewer #2: Yes

Reviewer #3: Yes

4. Is the manuscript presented in an intelligible fashion and written in standard English?

Reviewer #1: Yes

Reviewer #2: Yes

Reviewer #3: Yes

5. Review Comments to the Author

Reviewer #1: In this single-center retrospective cohort study of 150 pediatric burn patients with severe oil burns, the authors evaluated the relationship between hospitalization duration and albumin, CRP, and CRP-albumin ratio values at time of admission. The results suggested that lower albumin levels, higher CRP levels, and higher CRP-albumin ratios at time of admission were associated with longer hospitalizations. The authors’ aim is an interesting question and one that could be readily applied at the bedside given that these are commonly obtained values. However, this manuscript has several major weaknesses. It requires significant copyediting and has multiple grammatical errors. The only outcome measure that was analyzed in this study was duration of hospitalization, which was used to extrapolate predictions about “prognosis.” Fortunately, there were few deaths in their database, but these patients were excluded which further limits the study and its ability to evaluate eventual outcomes given that mortality was excluded. The study only included oil burns, while excluding thermal and electrical burns or patients with inhalational injuries, which limits generalizability. Additionally, as mentioned in the discussion, a limitation was that burn area localization was not accounted for; I would add that there was no mention of total body surface area which is an important factor in severity of burns. Overall, the conclusions drawn in lines 98-100 stating that the study showed that clinical prognosis in the pediatric burn population can be predicted by the CRP-albumin ratio feels overstated.

The abstract overall does a good job summarizing the key points of the article. The results section here does feel slightly redundant.

Overall, I think this introduction would be a lot stronger if the interesting pathophysiology explanations in the discussion section were moved to the introduction. See my comment #12 below.

Line 24: consider rewording this sentence without the use of the word “important”

Line 29: consider being more specific about timeframe of CRP increase instead of “increases in a short time”

Lines 32-33: utilization of word “important” twice in this sentence

Lines 39 and 48: instead of “third-degree and above severe burns,” consider rephrasing to “third- and fourth-degree burns”

Line 47: I am a little confused about the inclusion of patients who were treated at your medical center’s burn clinic. Were they all admitted from clinic?

Lines 54-55: I would be interested to hear more about the data regarding patients who died and why it was excluded. What was the number of patients?

Lines 56-57: Why were these groups excluded? Curious about why the focus on oil burns only and why inhalation burns were excluded.

Line 76-77: This could be moved to the “Patients and data” section.

Lines 78-89: In the results section, I would have liked to have seen the numerical lab values included in parentheses for the second and third paragraphs. These read more like a discussion section.

Lines 101-107, 115-117, 125-131: This is all very interesting and well-researched background that I think belong in the introduction section.

Lines 139-142: This sentence is copied word-for-word directly from the results section.

Lines 146-149: This sentence is difficult to follow initially because “these patients” are referenced before the group (pediatric burn patients who died) is specified.

Reviewer #2: The manuscript document laboratory data as prognostic guide

It would be better to have more information on the clinical course

Including antibiotic treatment, nutritional status and treatment.

Albumin support as needed and skin graft.

Reviewer #3: 1. Pediatric patients who died due to burns were excluded from the study because of their low number. This limits the ability to analyze whether CRP-albumin ratio and other markers could predict mortality, which is an important outcome in severe burn cases.

2. The study did not evaluate patients based on burn area localization. This is a significant limitation, as the location of burns can influence healing, complications, and hospitalization duration.

3. Patients with inhalation burns were excluded from the study. These burns often have a distinct clinical course and outcomes, and excluding them may limit the applicability of findings to all pediatric burn patients.

4. The study focuses on patients treated at a single hospital, which may limit the generalizability of findings to other regions, healthcare systems, or patient populations.

5. The threshold value for the CRP-albumin ratio was derived from the study's ROC analysis. This may lead to overfitting and a need for external validation in larger, independent cohorts to confirm the threshold's applicability.

6. The retrospective nature of the study introduces potential biases, such as incomplete data collection or variability in clinical practices over the study period.

7. The study only analyzed CRP, albumin, and CRP-albumin ratio at admission. Longitudinal changes in these markers could provide more detailed insights into their relationship with the clinical course and outcomes.

8. While the mean age and gender distribution are provided, other demographic or socioeconomic factors that might influence recovery or treatment outcomes are not discussed.

9. The sensitivity (75%) and specificity (61.2%) of the CRP-albumin ratio for predicting hospitalization for more than 20 days are moderate, indicating that the marker alone may not be highly reliable without additional context.

10. Although the sample size calculation ensured adequate power, the exclusion of certain subgroups (e.g., older patients and those with specific comorbidities) may limit the study's scope.

6. PLOS authors have the option to publish the peer review history of their article (what does this mean? ). If published, this will include your full peer review and any attached files.

**Do you want your identity to be public for this peer review?** For information about this choice, including consent withdrawal, please see our Privacy Policy .

Reviewer #1: No

Reviewer #2: No

Reviewer #3: No

---

## [Author Response · Author response to Decision Letter 0]

1 Mar 2025

Dear Editor,

We thank the all reviewers' for evaluating our article. The all reviewers' responses were answered point by point. The file containing the comments is uploaded as an attachment.

Kind regards

---

## [Editor Report · Decision Letter 1]

3 Mar 2025

Relationship Between C-reactive Protein-Albumin Ratio and Prognosis in Pediatric Patients with Severe Burns

PONE-D-24-38904R1

Dear Dr. Görgün,

We’re pleased to inform you that your manuscript has been judged scientifically suitable for publication and will be formally accepted for publication once it meets all outstanding technical requirements.

Kind regards,

Keiko Hosohata, Ph.D.

Academic Editor

PLOS ONE
---

## [Editor Report · Acceptance letter]

PONE-D-24-38904R1

PLOS ONE

Dear Dr. Görgün,

I'm pleased to inform you that your manuscript has been deemed suitable for publication in PLOS ONE. Congratulations! Your manuscript is now being handed over to our production team.

Kind regards,

on behalf of

Dr Keiko Hosohata

Academic Editor

PLOS ONE